# 🦙MINER: Mining the Underlying Pattern of Modality-Specific Neurons in Multimodal Large Language Models

## Abstract

In recent years, multimodal large language models (MLLMs) have significantly advanced, integrating more modalities into diverse applications. However, the lack of explainability remains a major barrier to their use in scenarios requiring decision transparency. Current neuron-level explanation paradigms mainly focus on knowledge localization or language- and domain-specific analyses, leaving the exploration of multimodality largely unaddressed. To tackle these challenges, we propose **MINER**, a transferable framework for **mi**ning modality-specific **neur**ons (MSNs) in MLLMs, which comprises four stages: ❶ modality separation, ❷ importance score calculation, ❸ importance score aggregation, ❹ modality-specific neuron selection. Extensive experiments across six benchmarks and two MLLMs show that **(I)** deactivating ONLY 2% of MSNs significantly reduce MLLMs performance ($0.56 \sim 0.24 \downarrow$ for Qwen2-VL, $0.69 \sim 0.31 \downarrow$ for Qwen2-Audio), **(II)** different modalities mainly converge in the lower layers, **(III)** MSNs influence how key information from various modalities converges to the last token, **(IV)** We observed two intriguing phenomena, semantic probing and semantic telomeres.

## 1 Introduction

Recently, multimodal language models (MLLMs) have made rapid advancements across various applications (Xu et al., 2024; Xiao et al., 2024; Yan et al., 2024), exemplified by models like GPT-4 (Achiam et al., 2023), LLaMA 3 (Dubey et al., 2024), Qwen-VL (Bai et al., 2023b), and LLaVA-NEXT (Liu et al., 2024a). However, their

Table 1: X-specific neuron studies.

| | | **X-Specific Neurons** | | |
| --- | --- | --- | --- | --- |
| | | Language
Tang et al. (2024a) | Domain
Huo et al. (2024) | **Modality**
**Ours** 🦙 |
| **Granularity** | Sample-level | ✔ | ✔ | ✗ |
| | Token-level | ✗ | ✗ | ✔ |
| **Importance Metrics** | Probability | ✔ | ✔ | ✔ |
| | Mean | ✗ | ✗ | ✔ |
| | Max | ✗ | ✗ | ✔ |
| | Attention | ✗ | ✗ | ✔ |

*black-box* nature presents challenges, particularly in fields like medical studies (González-Alday et al., 2023), where interpretability is essential. Understanding the decision-making process is vital, making explainability a central focus of ongoing research (Tjoa & Guan, 2020; Zhao et al., 2024).

Numerous studies have sought to understand how knowledge is stored in models (Sukhbaatar et al., 2019; Dai et al., 2021; Meng et al., 2022a; Chen et al., 2024a) and how this information influences decision-making (Geva et al., 2020; Petroni et al., 2019). For example, Dai et al. (2021); Geva et al. (2020) investigate knowledge storage mechanisms, while Wendler et al. (2024); Zhang et al. (2024) provide insights into layer-level explainability. Additionally, several works in the neuron-level explanation domain (Tang et al., 2024a; Kojima et al., 2024; Huo et al., 2024) have identified language-specific or domain-specific neurons, referred to as *X-specific neurons*. However, these studies often neglect modality-level understanding, particularly how multimodal information is processed and its differences and similarities (Parekh et al., 2024; Rodis et al., 2023), as shown in table 1.

As shown in table 1 and fig. 1, recent X-specific neuron works (Tang et al., 2024a; Huo et al., 2024) face two key limitations: First, they *focus on sample-level neuron identification*, assuming each sample belongs to a single language or domain, while multimodal samples often span multiple

Figure 1: Comparison of Language-specific **(a)**, Domain-specific **(b)**, and our proposed Modality-specific Neuron detection and analysis framework, MINER **(c)**.

modalities. Second, they *rely solely on activation probability as the importance metric*, which is insufficiently comprehensive (See section 4.3 and Ob2 of section 5.4 for details).

This raises an intriguing question: *Are there modality-specific neurons (MSNs) akin to those in multi-language or multi-domain settings?* To address this, we must tackle the following challenges: ❶ How can we measure neuron importance for specific modalities? ❷ What mechanisms do these neurons use to influence the model? ❸ Can we identify some underlying patterns of MSNs?

We propose **MINER**, a transferable framework that develops new importance metrics for measuring neuron significance and introduces selection strategies for identifying key neurons for each modality. MINER tackles the challenges by: ❶ Decompose neuron importance for modalities into token importance from the top down, then restore neuron-modal importance through bottom-up aggregation. ❷ Analyze neuron behavior using feature dimensionality reduction plots and contribution scores of modality tokens to predictions. ❸ Our experiments revealed semantic probing and semantic telomeres. Through this design, we identified a set of important modality-related neurons and uncovered interesting phenomena during our experimental analysis, offering valuable insights for the research community. Our contributions can be summarized as follows:

☞ To our knowledge, we are the first to analyze modality-specific neurons (MSNs) in multimodal large language models (MLLMs).

☞ We introduce MINER, a transferable framework for selecting MSNs in both vision-based and audio-based MLLMs, capable of handling datasets with any combination of modalities uniformly.

☞ We provide a systematic problem definition and propose a novel token-level analysis pipeline that differs from existing sample-level methods.

☞ We validate the significance of MSNs through extensive experiments, uncover intriguing phenomena of semantic probing and semantic telomeres, and present new insights.

## 2 RELATED WORKS

In this section, we provide a brief overview of knowledge location and neuron analysis work, while the related work for the remaining sections is included in appendix B.1.

**Knowledge Localization.** Research has examined how factual and commonsense knowledge is represented in neural networks (Park et al., 2024; Hase et al., 2024; Zhu et al., 2024). For instance, Sukhbaatar et al. (2019) demonstrated that persistent memory vectors can replace feed-forward network (FFN) layers in transformers without performance loss. Geva et al. (2020) showed that FFN layers serve as key-value memories, linking training patterns to output vocabulary. Recently, Dai et al. (2021) identified *knowledge neurons* in FFN layers of pretrained transformers, positively correlating their activation with factual knowledge expression. Meng et al. (2022a) and Chen et al. (2024a) further explored knowledge neurons in language models. In this context, our analysis aims to identify modality-specific neurons in the FFN layers of MLLMs.

**Neuron Analysis.** Neuron analysis in pretrained models is an emerging research area in both computer vision and natural language processing. Recent studies have gone beyond explaining the concepts and knowledge represented by individual neurons (Bau et al., 2017; Oikarinen & Weng, 2022; Bills et al., 2023; Gao et al., 2024) to identify neurons that respond to specific patterns. For example, Schubert et al. (2021) and Cammarata et al. (2020) found visual neurons that detect high-frequency features or curves in images, while Tang et al. (2024a) and Kojima et al. (2024) analyzed neurons uniquely activated by target languages, termed *language-specific neurons*, in large language models. In the multimodal domain, research has primarily focused on detecting neurons that respond to both textual and visual inputs (Goh et al., 2021; Schwettmann et al., 2023; Pan et al., 2023). Additionally, Huo et al. (2024) adapted techniques from Tang et al. (2024a) to identify domain-specific neurons (e.g., in medicine and remote sensing) in MLLMs. However, the specialization of neurons in multimodal models is still underexplored. To our knowledge, our work is the **first** to analyze modality-specific neurons in MLLMs.

# 3 PRELIMINARIES

## 3.1 DEFINITIONS OF MODALITY, SAMPLE AND DATASET

We start by clarifying two key terms: modality set and modality space, which will be referenced throughout the paper. Building on these definitions, we then define samples and datasets to enhance the understanding of our method.

**Modality Set.** The classification of modalities in data is not singular. Here, we outline three potential methods for partitioning modality sets:

$$S_{\text{all}} = \{\texttt{all}\}$$
$$S_M = \{\texttt{text}, \texttt{special}, \texttt{image}, \texttt{video}, \texttt{audio}\} = \{m_1, \dots, m_M\}$$
$$S_{\text{t+s}} = \{\texttt{t+s}, \texttt{image}, \texttt{video}, \texttt{audio}\}$$

We employ $S_M$ in our method, while $S_{\text{all}}$ (treats all as one modality) and $S_{\text{t+s}}$ are used **only** in ablation studies. The **special** modality refers to elements not present in the raw data but introduced by certain MLLMs. For instance, in Qwen2-VL, this includes separators like [im-start] and [im-end] or placeholders such as [image-pad]. The **only** difference between $S_M$ and $S_{\text{t+s}}$ is whether we consider **special** and **text** as the same modality. If treated as the same, we derive $S_{\text{t+s}}$.

**Modality Space.** Each modality $m \in S_M$ corresponds to a modality space $\mathcal{X}_m$, encompassing all data or features associated with that modality. This can be formally defined as:

$$\mathcal{X}_m = \{x_m \mid x_m \text{ is a feature corresponding to modality } m_i \text{ from a multimodal sample}\} \quad (1)$$

Additionally, modality space can be viewed as a collection of exclusive information, where data from one space should not overlap with others. For example, the image modality space $\mathcal{X}_{\text{image}}$ includes raw images and features, encompassing all samples related to visual information.

**Multimodal Space / Sample.** With the foundational definitions of modality established, we define the multimodal sample space as $\mathcal{X}$, where a sample is represented as:

$$\mathbf{x} = (x_m \mid m \in S_M) \quad \text{where} \quad x_m = \begin{cases} \text{actual modality data} & \text{if modality } m \text{ exists} \\ \text{None} & \text{if modality } m \text{ does not exist} \end{cases}$$

where $x_m \sim \mathcal{X}_m$. For simplicity, we omit $x_m = \text{None}$, allowing a VQA sample to be expressed as $\mathbf{x} = (x_{\text{text}}, x_{\text{image}})$. To perform a fine-grained analysis of multimodal samples, we define the modality function Mod to extract the modality of component $x_m$ as follows:

$$\text{Mod}(x_m) = \begin{cases} m & \text{if } x_m \neq \text{None} \\ \varnothing & \text{otherwise} \end{cases}$$

The modality function can be generalized to extract a sample's modalities, returning a set of modalities: $\text{Mod}(\mathbf{x}) = \{\text{Mod}(x_m) \mid m \in S_M\}$. For example, $\text{Mod}(\mathbf{x} = (x_{\text{text}}, x_{\text{image}})) = \{\texttt{text}$. We divide the sample space $\mathcal{X}$ into two mutually exclusive subspaces based on the number of modalities: the uni-modality space $\mathcal{X}_{\text{uni}} = \{\mathbf{x} \mid \#\{x_i \mid x_i \neq \text{None}\} = 1\}$ and the multi-modality space $\mathcal{X}_{\text{multi}} = \{\mathbf{x} \mid \#\{x_i \mid x_i \neq \text{None}\} > 1\}$.

**Multimodal Dataset.** A dataset is a subset of $\mathcal{X}$ where all samples share common characteristics, such as modality or question type. For example, VQA datasets contain both text and image modalities, structured around answering questions. We define uni-modality datasets as $D^{\text{uni}} = \{\mathbf{x}_1, \mathbf{x}_2, \dots \mid \mathbf{x}_i \in \mathcal{X}_{\text{uni}}\}$, and multi-modality datasets as $D^{\text{multi}} = \{\mathbf{x}_1, \mathbf{x}_2, \dots \mid \mathbf{x}_i \in \mathcal{X}_{\text{multi}}\}$.

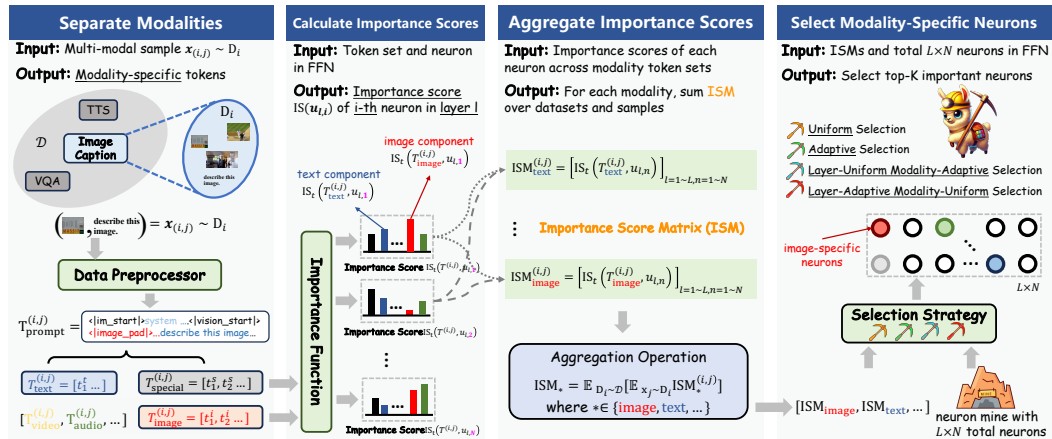

Figure 2: **Four stages of MINER. ❶** Prompt tokens are divided into modality sets before being input into the LLM. **❷** Each neuron computes an importance score for the tokens of each modality. **❸** Aggregate these values to compute the Importance Score Matrix (ISM), reflecting the modality-level importance of each neuron. **❹** Various selection methods, as detailed in section 4.5, are employed to identify modality-specific neurons for each modality.

## 3.2 RELEVANT TRANSFORMER CONCEPTS

MLLMs process **x** using modality-specific encoders and tokenizers (e.g., ViT (Dosovitskiy, 2020a) for **image**), transforming it into a set of **input tokens**:

$$T_{\text{input}} = \bigcup_{m \in S_M} T_m = \{t_1, t_2, \ldots, t_I\} \quad \text{where} \quad T_m \cap T_{m'} = \emptyset \quad \text{for } m \neq m' \tag{2}$$

By extending Mod to the token domain, we define $T_m = \{t_i \in T_{\text{input}} | \text{Mod}(t_i) = m\}$ as the modality-specific token set, with size $|T_m| = I_m$. We also establish a mapping structure $\text{Ind}_m$ for each subset, enabling retrieval of the original index from the input token set. This allows us to trace any element in $T_m$ back to the original set using the relationship $T_{\text{input}}[\text{Ind}_m[i]] = T_m[i]$.

To enhance clarity in the subsequent definitions, we represent the token embedding corresponding to $t_i$ after being input into the LLM at layer $l$ as $t_i^l \in \mathbb{R}^d$, without distinguishing between the token and its embedding. We denote the embeddings at layer $l$ as $T_{\text{input}}^l = [t_i^l]_{i=1}^I \in \mathbb{R}^{I \times d}$ and the attention matrix after softmax as $\mathbf{A}^l \in \mathbb{R}^{I \times I}$. Ignoring layer normalization, we denote the $i$-th value vector as $v_i^l$, allowing us to express the embedding update process as:

$$a_i^l = t_i^l + \sum_{j=1}^{I} \mathbf{A}_{i,j}^l v_j^l, \quad t_i^{l+1} = a_i^l + W_{\text{out}}^l \text{Act}(W_{\text{in}}^l a_i^l) \tag{3}$$

where $W_{\text{in}} \in \mathbb{R}^{d \times N}$ and $W_{\text{out}} \in \mathbb{R}^{N \times d}$ is the up-sampling and down-sampling layers, respectively. In the equation above, $a_i^l$ represents the output of the attention module, and $\mathbf{H}^l \in \mathbb{R}^{I \times N}$ is the hidden activation vector ($\mathbf{H}_{i,n}^l$ represents the activation value of $n$-th neuron for token $t_i$). Each neuron is denoted as $u_{l,n}$, resulting in a total of $L \times N$ neurons in the FFN modules, represented as the matrix $U_{L \times N}$. Then these embeddings pass through $L$ identical transformer blocks, generating output tokens sequentially in an autoregressive manner. We define the set of **output tokens** as $T_{\text{output}} = \{t_{I+1}, t_{I+2}, \ldots, t_{I+O}\}$.

## 4 FRAMEWORK OF MINER

### 4.1 PROBLEM ANALYSIS

Motivated by previous research on X-specific neurons, such as language-specific Tang et al. (2024a) and domain-specific neurons Huo et al. (2024), our work focuses on identifying a set of MSNs that are *critical for processing multimodal samples in MLLMs*. We briefly analyze how our approach differs from previous studies in fig. 1 and table 1. We focus on neurons in the FFN of MLLMs rather

than other modules, such as modality-specific encoders or projection layers, for several reasons: (**I**) Previous studies indicate that FFN encode distinct, recoverable knowledge attributes (Geva et al., 2020; Dai et al., 2021; Meng et al., 2022a;b), with some neurons representing the same concepts across different modalities (Schwettmann et al., 2023). (**II**) Research by Schwettmann et al. (2023) identifies semantic alignment between images and text within the LLM, rather than in the projection layer. (**III**) Our goal is to identify neurons across modalities within the same set, so we exclude modality-specific encoders. Thus, our research focuses on the FFN module.

Our problem can be distilled into two steps: first, measure a neuron's importance for a specific modality (Stages 1-3), and second, select the most important neurons based on this measure (Stage 4). Assessing the importance of $u_{l,n}$ in modality space $\mathcal{X}$ is complex due to the infinite number of potential samples. Therefore, we define the **importance score** between a neuron and samples as $\text{IS}_s(u_{l,n}, \mathbf{x})$. If a neuron consistently demonstrates importance across many samples, we can conclude it is significant within the modality space. The four stages of MINER align with the sections in fig. 2, outlined as follows:

➠ **Stage 1: Modality Separation.** Decomposing the information within the LLM by modality.

➠ **Stage 2: Importance Score Calculation.** We decompose the importance of neurons to modalities into their token-level significance and define it accordingly.

➠ **Stage 3: Importance Score Aggregation.** We aggregate the token-level importance of neurons to restore their modality-level significance.

➠ **Stage 4: Modality-Specific Neuron Selection.** The aggregated scores and a selection strategy are used to identify the top-$K$ important MSNs.

### 4.2 Stage 1: Modality Separation

To identify neurons associated with specific modalities, we first need to separate the information by modality. While input tokens $T_{\text{input}}$ can be divided into distinct sets $\{T_{\text{image}}, T_{\text{text}}, \ldots\}$, the attention mechanism blends information across modalities, complicating complete separation. Inspired by the VQA demo in fig. 3, which shows that a small portion of license plate information suffices to answer the question, we propose the following hypothesis:

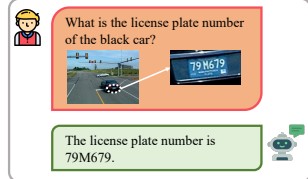

Figure 3: A VQA demo.

**Hypothesis 1:** *As information passes through the LLM layers, most of the content in $T_{m_i}$ remains within its set, with only a small portion related to the question being transferred.*

This hypothesis is supported by confirmatory experiments in section 5.5, prompting us to propose an approximate method for segmenting modality information: we assume the attention module functions only within distinct token sets of different modalities, preventing information exchange between subsets and enabling the partitioning of tokens into mutually exclusive modalities.

Our hypothesis generates a set of uni-modality datasets $\{D^1_{\text{uni}}, D^2_{\text{uni}}, \ldots\}$ at the **token level** by partially limiting information flow, aligning to some extent with Tang et al. (2024a); Huo et al. (2024). We further refine the neuron-sample importance function $\text{IS}_s$ by defining it at a more granular level between neuron and token set as $\text{IS}_t(u_{l,n}, T_m)$. The next stage outlines how we utilize modality information and provides a detailed definition of the importance score.

### 4.3 Stage 2: Importance Score Calculation

In the FFN, each neuron outputs an activation value for a token, which naturally serves as an importance score and is widely used (Tang et al., 2024a; Huo et al., 2024). We use $\mathbf{H}^l_{i,n}$, the activation value introduced in section 3.2, as the baseline for all other importance scores (sample-level $\text{IS}_s$, and token-level $\text{IS}_t$). As shown in fig. 12, the significant variation in token counts across modalities makes individual token-based importance unreliable. Therefore, we calculate importance over a token set, removing token count as a factor. We propose the following operations to aggregate activation values for a token set:

➢ **Prob.** The activation value reflects a neuron's interest in a token, so we calculate the *interest probability* (activation value > 0) for the tokens:

$$\text{IS}_t^{\text{Prob}}(u_{l,n}, T_m) = \text{IS}_t^{\text{Prob}}(\mathbf{H}_{1,n}^l, \ldots, \mathbf{H}_{I_m,n}^l) = \frac{1}{I_m} \sum_{i=1}^{I_m} \mathbb{I}(\mathbf{H}_{i,n}^l > 0) \tag{4}$$

Here, $\mathbb{I}$ is the indicator function. A higher probability suggests greater engagement with the token set. However, high activation probability doesn't necessarily imply importance. A neuron may have frequent low activations, while fewer high values could indicate a greater contribution. These limitations are confirmed in section 5.4).

➢ **Mean.** Taking the average is a common approach that captures the group's overall characteristics while balancing the influence of individual tokens. We define this operation as follows:

$$\text{IS}_t^{\text{Mean}}(u_{l,n}, T_m) = \text{IS}_t^{\text{Mean}}(\mathbf{H}_{1,n}^l, \ldots, \mathbf{H}_{I_m,n}^l) = \frac{1}{I_m} \sum_{i=1}^{I_m} \mathbf{H}_{i,n}^l \tag{5}$$

➢ **Max.** We use the maximum activation value as an importance measure. Unlike $\text{IS}_t^{\text{Mean}}$, which represents a uniform distribution, $\text{IS}_t^{\text{Max}}$ corresponds to a Dirac distribution.

➢ **Attn-Q.** We developed a method that uses attention values to assess a neuron's importance for a token set, considering each $t_i \in T_{\text{input}}$:

$$\text{IS}_t(u_{l,n}, T_m) = \frac{1}{T} \sum_{i=1}^{T} \sum_{j=1}^{I_m} w_j(t_i) \mathbf{H}_{\text{Ind}_m[j],n}^l \tag{6}$$

We use $\text{Ind}_m$ to map the index of $T_m$ back to $T_{\text{input}}$, retrieving the corresponding values from $\mathbf{H}$. The $\text{IS}_t^{\text{Attn-Q}}$ employs the $i$-th row of the attention matrix $\mathbf{A}$, reflecting the attention scores for the $i$-th query across all keys, where $\mathbf{w}(t_i) = [w_j(t_i)]_{j=1}^{I_m} = \text{softmax}[\mathbf{A}_{i,\text{Ind}_m[j]}]_{j=1}^{I_m}$.

➢ **Attn-K.** Like $\text{IS}_t^{\text{Attn-Q}}$, $\text{IS}_t^{\text{Attn-K}}$ uses the $i$-th column of the attention matrix, reflecting the attention scores for the $i$-th key across all queries, where $\mathbf{w}(t_i) = \text{softmax}[\mathbf{A}_{\text{Ind}_m[j],i}]_{j=1}^{I_m}$.

As shown in table 1, previous studies on X-specific neuron analysis (Tang et al., 2024a; Huo et al., 2024) rely solely on $\text{IS}^{\text{Prob}}$, which has inherent limitations. To address this, we define $\text{IS}_t$ as the weighted sum of the five metrics discussed earlier, combining both local and global perspectives. We also evaluate the effectiveness of each metric in section 5.4.

### 4.4 STAGE 3: IMPORTANCE SCORE AGGREGATION

We calculated the importance score for each neuron across all modality-specific token sets. Next, we aggregate these scores to define the sample-level importance score:

$$\text{IS}_s(u_{l,n}, \mathbf{x}) = \text{IS}_s(u_{l,n}, T_{\text{input}}) = [\text{IS}_t(u_{l,n}, T_m)]_{m \in S_M} \in \mathbb{R}^M \tag{7}$$

For each neuron, we compute a vector as described above, resulting in a sample-level Importance Score Matrix (ISM) defined as $\text{ISM}_s(\mathbf{x}) = [\text{IS}_s(u_{l,n}, \mathbf{x})]_{L,N} \in \mathbb{R}^{M \times L \times N}$. We then aggregate the importance scores across samples from different datasets to obtain the modality-level ISM:

$$\text{ISM}(\mathcal{X}) = \sum_i \sum_j^{|D_i^{\text{multi}}|} \text{ISM}_s(\mathbf{x}_{i,j}) \in \mathbb{R}^{M \times L \times N}, \text{ where } \mathbf{x}_{i,j} \text{ is the } j\text{-th sample of } D_i^{\text{multi}} \tag{8}$$

As the sample size grows, the importance matrix becomes more effective in assessing neuron significance across the entire modality space.

### 4.5 STAGE 4: MODALITY-SPECIFIC NEURON SELECTION

The previous stage's $\text{ISM}(\mathcal{X})$ evaluates the importance of all neurons in the MLLMs for each modality. We then designed four strategies to select the $K$ highest importance scores from the ISM matrix. However, because of potential neuron overlap, the final number of selected neurons may be less than $K$. The strategies are implemented as follows:

**(I) Uniform** selects $\lfloor \frac{K}{L \times M} \rfloor$ neurons for each modality in every layer (the strongest assumption).

**(II) LA-MU** (Layer-Adaptive & Modality-Uniform) selects $\lfloor \frac{K}{M} \rfloor$ neurons for each modality, allowing for adaptive quantities in each layer and relaxing certain constraints.

**(III) LU-MA** (Layer-Uniform & Modality-Adaptive) selects $\lfloor \frac{K}{L} \rfloor$ neurons for each layer.

**(IV) Adaptive** selects $K$ positions in the ISM matrix without constraints on modality or layer.

fig. 4 intuitively illustrates the segmentation methods for the four selection strategies. The assumptions range from strong to weak, beginning with a fixed number of neurons for each modality in every layer and progressively relaxing constraints until the fourth approach imposes no limitations. Together, these strategies offer a comprehensive set of selection criteria.

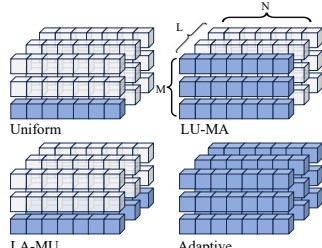

Using any selection strategy, we generate neuron positions as a Boolean mask $\mathbf{B} \in \{0,1\}^{L \times N}$ (with 1 indicating MSNs). As shown in eq. (8), different datasets yield distinct ISM, resulting in varied masks. Thus, with all other settings fixed, a specific dataset combination uniquely determines a neuron mask. For simplicity, we can define a mask function as $\mathbf{B} = \text{mask}(D_1, D_2, \ldots)$.

Figure 4: Selection strategies.

## 5 EXPERIMENT

We apply MINER in various settings to explore the existence and characteristics of MSNs. Our study aims to answer the following research questions:

✦ RQ1: Do the identified modality-specific neurons significantly contribute to multimodal models?

✦ RQ2: If RQ1 is validated, how do these neurons facilitate this contribution?

✦ RQ3: How do different hyperparameter settings influence the behavior of the MLLMs?

✦ RQ4: Can we uncover underlying patterns among modality-specific neurons?

### 5.1 EXPERIMENT SETUP

Unless otherwise noted (e.g., in ablation studies), we adopt the following default settings. We define $S_M$ as the modality set, treating `special` and `text` as distinct modalities. We deactivate neurons across **all modalities** by setting their outputs to zero, then evaluate the impact on performance.

**Models.** We select Qwen2-VL (Wang et al., 2024) for visual tasks and Qwen2-Audio (Chu et al., 2024) for audio tasks to ensure a thorough exploration of the current model landscape. Qwen2-VL processes text, image, and video modalities, while Qwen2-Audio focuses on text and audio.

**Datasets.** For text-only tasks, we chose MMLU (Hendrycks et al., 2020). For text-image tasks, we selected TextVQA (Singh et al., 2019), and for text-video tasks, we opted for MSVD-QA (Chen & Dolan, 2011). Since no datasets exist for image-only or audio-only tasks, we selected specific datasets with fixed prompts to minimize the impact of the text modality. For an approximate image-only dataset, we utilized the MS-COCO 2014 captioning benchmark (Lin et al., 2014), adopting the Karpathy split test set as per Li et al. (2023); Tang et al. (2024b); Zhan et al. (2024) by fixing the text tokens. For an approximate audio-only dataset, we employed LibriSpeech (Panayotov et al., 2015) and VocalSound (Gong et al., 2022) in a similar manner.

### 5.2 MAIN RESULTS (RQ1)

To address RQ1, we generate neuron masks for each dataset combination to thoroughly evaluate our method. The main results for Qwen2-VL and Qwen2-Audio are shown in table 2 and table 3. Key experimental observations (**Obs**) include:

**Ob1. Masking just 2% of neurons impacts performance.** The first row in both tables shows normal inference without masking. After masking the selected neurons, we

Table 3: **Qwen2-Audio results:** Same format as table 2.

| DATASETS | | | QWEN2-AUDIO ($0.69 \sim 0.31 \downarrow$) | | | |
|---|---|---|---|---|---|---|
| | | | **MMLU** | **LibriSpeech** | **VocalSound** | **Average** |
| **MMLU** | **Libri** | **Vocal** | **Accuracy** | **WRR** | **Accuracy** | |
| - | - | - | 0.40 | 0.94 | 0.74 | 0.69 |
| ✓ | | | 0.00 | 0.53 | 0.41 | 0.31 |
| | ✓ | | 0.01 | 0.85 | 0.15 | 0.34 |
| | | ✓ | 0.01 | 0.87 | 0.19 | 0.36 |
| ✓ | ✓ | | 0.00 | 0.85 | 0.34 | 0.40 |
| ✓ | | ✓ | 0.01 | 0.86 | 0.37 | 0.39 |
| | ✓ | ✓ | 0.01 | 0.81 | 0.35 | 0.39 |
| ✓ | ✓ | ✓ | 0.01 | 0.83 | 0.40 | 0.41 |

observe performance drops, except for a slight increase in MSVD (analyzed later). Results in table 4 show that randomly masking 2% of neurons has no effect, confirming that the modality-specific neurons identified by our method significantly influence model performance.

Table 2: **Main results of Qwen2-VL.** We select MSNs (2%) using various dataset combinations (indicated by ✓) and then mask neurons across all modalities, recording the performance of the masked MLLMs. The minimum value in each column is marked in blue. We highlight any new values in gray when adding a new dataset improves the mask's quality.

| DATASETS | | | | QWEN2-VL (0.56 ∼ 0.24 ↓) | | | | | | |
|---|---|---|---|---|---|---|---|---|---|---|
| | | | | TextVQA | COCO Caption | | | MMLU | MSVD-QA | Average |
| TextVQA | COCO Caption | MMLU | MSVD-QA | Accuracy | BLEU | S-BERT | CIDEr | Accuracy | Accuracy | |
| - | - | - | - | 0.90 | 0.10 | 0.79 | 0.36 | 0.69 | 0.51 | 0.56 |
| ✓ | | | | 0.75 | 0.04 | 0.18 | 0.10 | 0.57 | 0.52 | 0.36 |
| | ✓ | | | 0.80 | 0.01 | 0.14 | 0.18 | 0.52 | 0.53 | 0.36 |
| | | ✓ | | 0.87 | 0.03 | 0.70 | 0.34 | 0.59 | 0.50 | 0.51 |
| | | | ✓ | 0.38 | 0.01 | 0.14 | 0.18 | 0.44 | 0.32 | 0.25 |
| ✓ | ✓ | | | 0.77 | 0.04 | 0.28 | 0.16 | 0.55 | 0.54 | 0.39 |
| ✓ | | ✓ | | 0.79 | 0.03 | 0.19 | 0.14 | 0.56 | 0.46 | 0.36 |
| ✓ | | | ✓ | 0.78 | 0.05 | 0.33 | 0.20 | 0.57 | 0.54 | 0.41 |
| | ✓ | ✓ | | 0.79 | 0.02 | 0.41 | 0.27 | 0.50 | 0.45 | 0.41 |
| | ✓ | | ✓ | 0.85 | 0.10 | 0.61 | 0.34 | 0.59 | 0.52 | 0.50 |
| | | ✓ | ✓ | 0.40 | 0.01 | 0.15 | 0.17 | 0.41 | 0.31 | 0.24 |
| ✓ | ✓ | ✓ | | 0.76 | 0.04 | 0.49 | 0.30 | 0.53 | 0.45 | 0.43 |
| ✓ | ✓ | | ✓ | 0.82 | 0.06 | 0.39 | 0.26 | 0.60 | 0.42 | 0.43 |
| ✓ | | ✓ | ✓ | 0.78 | 0.04 | 0.35 | 0.24 | 0.57 | 0.48 | 0.41 |
| | ✓ | ✓ | ✓ | 0.84 | 0.05 | 0.61 | 0.33 | 0.58 | 0.48 | 0.48 |
| ✓ | ✓ | ✓ | ✓ | 0.82 | 0.06 | 0.53 | 0.29 | 0.56 | 0.52 | 0.46 |

**Ob2. Neurons identified from diverse datasets are higher quality.** For MSVD-QA performance, apply mask(MSVD) drops performance to 0.32, while apply mask(MSVD, COCO) raises it to 0.52. This may be because the diverse text questions in MSVD-QA closely relate to video semantics, producing high-quality text-specific neurons. In contrast, the fixed text prompts in COCO Caption (see fig. 7) create a repetitive pattern, degrading neuron mask quality. MMLU's diverse text data causes a larger performance drop (0.31) when combined with MSVD-QA.

**Ob3. Adding more datasets can improve MSNs quality.** Many values in both tables are highlighted in Gray, showing that new datasets often enhance mask quality. However, some cases also demonstrate decreased effectiveness, possibly due to the diversity issues mentioned in **Ob3** or conflicts among dataset characteristics that lead to incompatible ISM matrices. We will investigate these phenomena in future work.

## 5.3 NEURON FUNCTIONALITY PRINCIPLES (RQ2)

This research question focuses on the intrinsic mechanisms of action of MSNs. Several key observations are as follows:

**Ob1. The audio modality exhibits a "semantic probing" effect towards the text modality, indicating a potential trend toward aligning key information across modalities.** We present the feature distribution under three masking settings in fig. 5-(a). The complementary mask is defined as $1 - \mathbf{B}$. As the layer depth increases, audio embeddings extend "tentacles" toward the embeddings of other modalities, a phenomenon we call "semantic probing." This suggests that the LLM aligns not the entire modal feature space, but specific key information.

**Ob2. A "semantic telomeres" phenomenon occurs in special modality, anchoring at the edges of the text modality's semantic space.** In fig. 5-(a), text embeddings initially form circular clusters that elongate into strips in deeper layers. Special tokens progressively align with and stabilize at the front of these clusters, a pattern we refer to as "semantic telomeres."

**Ob3. MSNs shape MLLMs behavior by directing how key information from different modality tokens converges into the last token.** We apply mask(COCO) to the COCO Caption dataset and calculate each modality's contribution score to the final prediction based on the accumulated attention between its token set and the last token. As shown in fig. 5-(b), applying a mask for a specific modality results in $\Delta < 0$, indicating reduced information flow to the last token.

## 5.4 ABLATION STUDIES (RQ3)

We present ablation studies on MINER's components and highlight key observations:

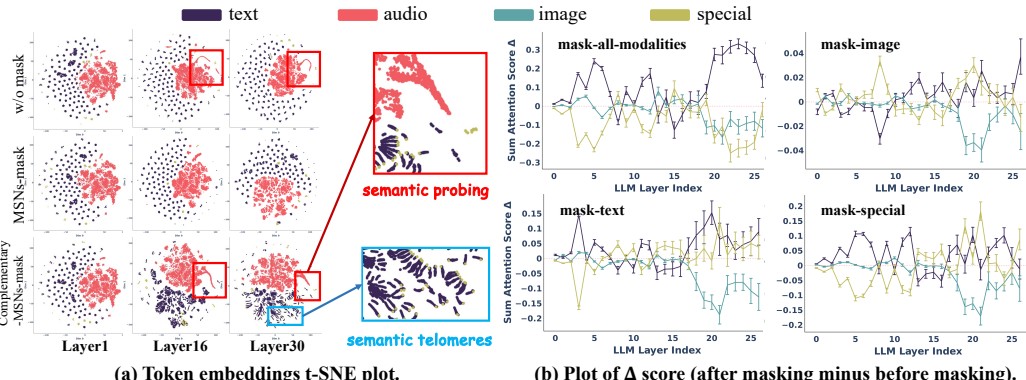

(a) Token embeddings t-SNE plot.

(b) Plot of Δ score (after masking minus before masking).

Figure 5: **(a)** t-SNE plots for VocalSound, showcasing three masking settings: no masking, Mask(VocalSound), and complementary masking (from top to bottom). **(b)** Display the change (Δ) in contribution scores between different token sets and the last token before and after masking, based on 100 samples (detail in section 5.3).

**Ob1. Selection strategies with appropriate degrees of freedom yield higher-quality MSNs.** As shown in table 4, the uniform method, which imposes a fixed number of neurons per modality at each layer, performed the worst. In contrast, greater flexibility in selecting neurons across modalities and layers enabled a better approximation of important neuron distribution. Among these strategies, LA-MU proved most effective, as it maintained flexibility in neuron counts across layers while treating all modalities equally, thus avoiding imbalances.

**Ob2. For the importance metrics, the activation probability was less effective than our other designs.** We evaluated the effectiveness of each of the five importance metrics individually and also assessed the impact of combining them. As shown in table 4, the activation probability performed poorly, aligning with the section 4.3 analysis. In contrast, our newly designed metrics, which account for both local and global perspectives, demonstrate improved effectiveness.

Table 4: **Ablation results.** Red (blue) indicates the minimum in each row (column), and green highlights values that are minimum in both.

| IMPORTANCE METRIC | | | | | SELECTION STRATEGY | | | | |
|---|---|---|---|---|---|---|---|---|---|
| Prob | Mean | Max | A-K | A-Q | Uniform | LU-MA | LA-MU | Adaptive | Random |
| 1 | | | | | 0.88 | 0.88 | 0.89 | 0.88 | 0.90 |
| | 1 | | | | 0.87 | 0.87 | 0.85 | 0.89 | 0.89 |
| | | 1 | | | 0.81 | 0.66 | 0.83 | 0.85 | 0.90 |
| | | | 1 | | 0.87 | 0.87 | 0.85 | 0.89 | 0.90 |
| | | | | 1 | 0.87 | 0.88 | 0.85 | 0.89 | 0.90 |
| | 1/2 | 1/2 | | | 0.83 | 0.82 | 0.81 | 0.17 | 0.90 |
| | | | 1/2 | 1/2 | 0.87 | 0.88 | 0.85 | 0.89 | 0.90 |
| | 1/4 | 1/4 | 1/2 | | 0.84 | 0.85 | 0.76 | 0.84 | 0.90 |
| | 1/4 | 1/4 | | 1/2 | 0.85 | 0.85 | 0.81 | 0.80 | 0.90 |
| 1/5 | 1/5 | 1/5 | 1/5 | 1/5 | 0.86 | 0.87 | 0.89 | 0.90 | 0.90 |

**Ob3. The model's performance significantly drops with an increase in masked MSNs.** As shown in fig. 6-(a), masking 1% of neurons has minimal impact, while 5% nearly collapses performance. Therefore, MINER selects 2% to strike a balance.

**Ob4. The model's performance is greatly influenced by the deactivation value settings.** Considering neuron $u_{l,n}$, we test three deactivation settings: fixing its output activation to 0, -0.1, and $\min(\mathbf{H}^l)$. As shown in fig. 6-(a), a slightly negative activation value reduces model performance, indirectly highlighting the importance of the identified neurons for the modality, since randomly deactivated neurons below zero do not impact performance.

**Ob5. Considering "special" and "text" as separate modalities leads to improved results.** We evaluate three modality sets: $S_{\text{all}}$, $S_M$ and $S_{\text{t+s}}$ defined in section 3.1, referred to as "all", "t,s", and "t+s" in fig. 6-(a). The poor performance of $S_{\text{all}}$ shows that our sample-level issue (all in one modality) cannot be resolved, highlighting the need for modality separation at the token level.

**Ob6. Masking MSNs from different modalities affects performance, with a greater impact as more modalities are masked.** We apply mask(COCO) to the COCO Caption dataset and normalize values within different metrics to a 0-1 range. Results are shown in Figure 1, with "t," "s," and "i" representing text, special, and image, respectively. We found that performance declines significantly with an increasing number of masked modalities.

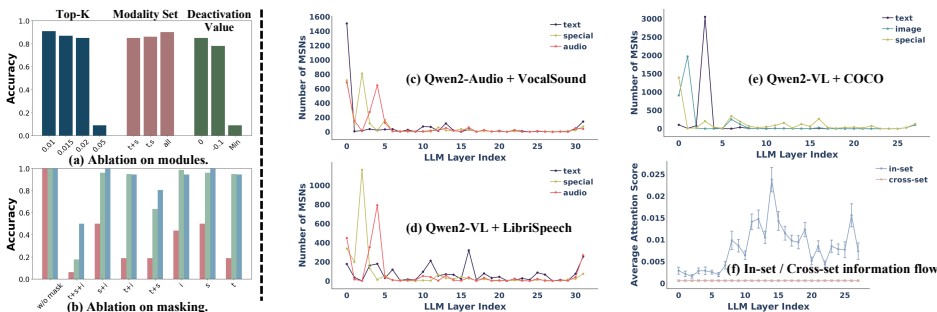

Figure 6: **(a)** and **(b)** present the ablation results, while **(c)**, **(d)**, and **(e)** illustrate the neuron distribution across layers. **(f)** shows the information flow within and between token sets.

## 5.5 IDENTIFYING PATTERNS IN NEURONS (RQ4)

In this section, we investigate potential MSNs patterns through experiments to offer valuable insights. Our key observations are as follows:

**Ob1. Most MSNs are concentrated in the shallow layers, with only a small portion in the final layers.** As shown in fig. 6, figures (c), (d), and (e) visualize the distribution of modality-specific neurons across layers. This trend aligns with findings in Zhang et al. (2024), indicating that cross-modal perception primarily occurs in the early layers (detailed in appendix B.2). Thus, we believe that different modalities primarily converge in the lower layers.

**Ob2. Most modality information stays within the token set, with only a small amount of key information transferring to other sets.** To validate Hypothesis 1 from section 4.2, we measure information flow using attention values, calculating the cumulative in-set attention and cross-set attention (the diagonal blocks of $\mathbf{A}$). We apply mask(MSVD-QA) to the MSVD-QA dataset, with results shown in fig. 6-(f). Our findings indicate that in-set information flow significantly surpasses cross-set flow, supporting our hypothesis.

## 6 CONCLUSION

To our knowledge, this is the **first** study of modality-specific neurons (MSNs) in MLLMs. We select a small set of key neurons from the FFN that are crucial for processing multimodal data and design experiments to investigate their underlying mechanisms. We address this issue through the following steps: ❶ Define key concepts related to modalities, samples, and datasets. ❷ Conduct sample-level and token-level analyses to differentiate modality information at a finer granularity, evaluate the limitations of existing importance metrics, and propose new ones. ❸ Calculate importance scores for neurons associated with a set of tokens and aggregate these scores across samples and datasets for a global modality-level analysis. ❹ Define four selection strategies to extract the top-$K$ neurons with the highest importance scores as the output MSNs. In the experimental section, we validate our method by masking the MSNs and observing the resulting decline in model performance, while also identifying two intriguing phenomena: semantic probing and semantic telomeres.

### 6.1 LIMITATION AND FUTURE WORK

Our work presents several areas for improvement: **(I)** We introduced the independence hypothesis of token sets to separate information by modality; however, key information exchanges between token sets occur and are difficult to capture. Future research could relax this assumption by incorporating cross-set information exchange for potentially better results. **(II)** Our study focused on vision-related Qwen2-VL and audio-related Qwen2-Audio models, limiting the range of modalities. Future work aims to include broader MLLMs, such as any-to-any models, and datasets with more diverse modalities. **(III)** We observed two intriguing phenomena—semantic probing and semantic telomeres—whose underlying causes remain unclear, presenting an opportunity for further exploration. **(IV)** The quality of datasets directly influences the quality of identified neurons, which may be linked to data diversity. Future research could investigate this further and design metrics to quantify dataset quality. **(V)** While this work emphasizes identifying important neurons, the next step is to explore how to leverage these neurons to enhance MLLMs performance on relevant modalities, potentially through neuron fine-tuning techniques.

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

# A DATASET DETAILS

This section presents a detailed description of the datasets used in our evaluation, including data samples and the prompts provided to MLLMs.

## A.1 TEXTVQA

TextVQA (Singh et al., 2019) is a dataset for Visual Question Answering (VQA) that focuses on answering questions about text found in images. It features a variety of images, such as signs and labels, requiring models to combine visual understanding with Optical Character Recognition (OCR) to accurately respond to questions. TextVQA is a key benchmark for evaluating the integration of visual and textual reasoning in AI models.

## A.2 COCO CAPTION

The COCO Caption dataset (Lin et al., 2014) is a large-scale resource containing over 1.5 million captions that describe more than 330,000 images, distributed across 80 diverse object categories. The images were selected to represent complex, real-world scenes, depicting everyday environments where common objects appear in their natural contexts. Each image is annotated with five captions, each of which was independently generated by human annotators. COCO Caption dataset has been a foundational benchmark for training and evaluating the image captioning performance of multimodal large language models.

## A.3 MMLU

The Massive Multitask Language Understanding (MMLU) (Hendrycks et al., 2020) dataset is a benchmark for evaluating language models across 57 subjects, ranging from elementary topics to advanced academic fields. It consists of multiple-choice questions testing a model's knowledge and reasoning across diverse domains. MMLU is widely used to assess the generalization capabilities of large language models.

## A.4 MSVD-QA

The MSVD (Microsoft Research Video Description Corpus) (Chen & Dolan, 2011), also known as YouTube2Text, consists of 1,970 short videos, each ranging from 10 to 25 seconds with an average duration of 9 seconds. These videos depict a variety of subjects, including people, animals, actions, and different scenes. Each video is annotated with multiple sentences by different annotators, averaging around 41 sentences per clip, resulting in a total of 80,839 sentences. On average, each sentence contains 8 words, with approximately 16,000 unique words across the dataset.

## A.5 LIBRISPEECH

LibriSpeech (Panayotov et al., 2015) is a widely used dataset for automatic speech recognition (ASR) tasks. It contains approximately 1,000 hours of 16kHz English speech, sourced from the LibriVox audiobooks. The dataset is divided into several subsets, including "clean" and "other", which distinguish recordings by noise levels. Each audio file is accompanied by an accurate transcription, making it ideal for training and evaluating ASR models. LibriSpeech's diverse speaker base and detailed annotations also make it suitable for tasks like speaker identification and voice synthesis.

## A.6 VOCALSOUND

VocalSound (Gong et al., 2022) is an open dataset containing 21,024 crowdsourced recordings of human vocalizations such as laughter, sighs, coughs, throat clearing, sneezes, and sniffing, from 3,365 individuals. Designed for classification tasks, it enables models to accurately identify various non-speech sounds. The dataset also includes metadata like speaker age, gender, native language, country, and health status, supporting research on how demographic factors affect vocal sounds.

With its comprehensive scope and detailed annotations, VocalSound is a valuable resource for improving models in human vocalization classification.

## A.7 PROMPTS OF DATASETS

Generate a caption for the image in one short sentence, similar to these examples from the COCO dataset:
1. A man with a red helmet on a small moped on a dirt road.
2. Man riding a motor bike on a dirt road on the countryside.
3. A man riding on the back of a motorcycle.
4. A man in a red shirt and a red hat is on a motorcycle on a hill side.

Now, describe the image.

Figure 7: Prompt for COCO Caption.

The following are multiple choice questions (with answers) about abstract algebra.

Find all c in $Z_3$ such that $Z_3[x]/(x^2 + c)$ is a field.
**A.** 0
**B.** 1
**C.** 2
**D.** 3
**Answer:** B
...

Find the degree for the given field extension Q(sqrt(2), sqrt(3), sqrt(18)) over Q.
**A.** 0
**B.** 4
**C.** 2
**D.** 6
**Answer:**

Figure 8: Prompt for MMLU.

Please transcribe the following audio directly into plain text without any additional explanations, prefixes, or descriptions.
Only output the transcription of the spoken content in the audio.

Figure 9: Prompt for LibriSpeech.

## A.8 DATASET SAMPLE STATISTICS

## B ADDITIONAL RESULTS

### B.1 ADDITIONAL RELATED WORKS

**Development of MLLMs.** Researchers have extensively investigated integrating additional modalities into foundational large language models (Liu et al., 2023). Notably, large vision-language models (Zhu et al., 2023) and audio-language models (Deshmukh et al., 2023) have gained significant

You are a sound classification model. Your task is to classify a given audio sample into one of the following categories based on its content:
**1.** Laughter
**2.** Sigh
**3.** Cough
**4.** Throat clearing
**5.** Sneeze
**6.** Sniff

Please analyze the audio sample and provide the corresponding category name.

Figure 10: Prompt for Vocal Sound.

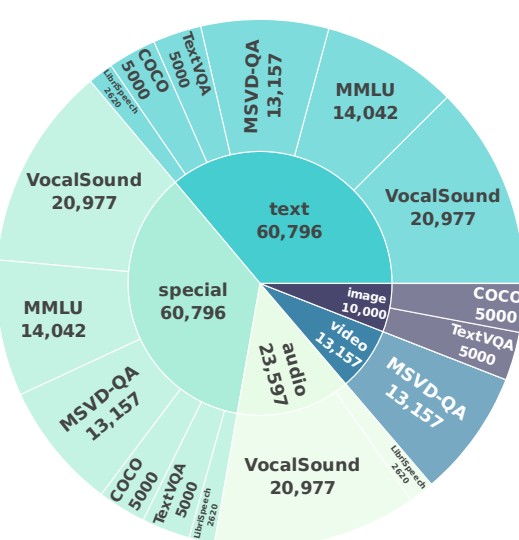

Figure 11: **Sample-level count statistics for different modalities.** We provide a sample-level statistical analysis of all datasets used in this work. The inner ring represents different modalities, while the outer ring corresponds to the respective datasets, with sample counts labeled on both rings. For instance, a VQA sample containing text, special, and image modalities is counted under each modality it includes.

attention by combining visual or audio inputs with text. For instance, (Liu et al., 2024b) proposed aligning images with text by projecting visual embeddings from a pretrained vision encoder into word space through a single MLP layer, allowing LLMs to understand the post-projection tokens. Similarly, Chen et al. (2024b) and Lu et al. (2024) used various projectors for this alignment. Recently, Qwen2-VL (Bai et al., 2023b) introduced a universal vision encoder that processes both images and videos, integrating visual embeddings directly into the textual token stream. Parallelly, several studies have focused on integrating audio data into LLMs (Deshmukh et al., 2023; Wu et al., 2023), typically involving post-processing of auditory embeddings through additional modules like Q-Former (Tang et al., 2023) or downsampling layers (Das et al., 2024). Notably, Qwen2-Audio (Chu et al., 2023) has surpassed previous state-of-the-art models across various audio benchmarks without task-specific fine-tuning. In this research, we selected Qwen2-VL and Qwen2-Audio as our vision-language and audio-language baselines, both utilizing Qwen-7B (Bai et al., 2023a; Yang et al., 2024) as the foundational LLM, with Vision Transformer (Dosovitskiy, 2020b) and Whisper-large-v3 (Radford et al., 2022) serving as their respective vision and audio encoders.

## B.2 ALL DISTRIBUTIONS OF NEURONS ACROSS LAYERS

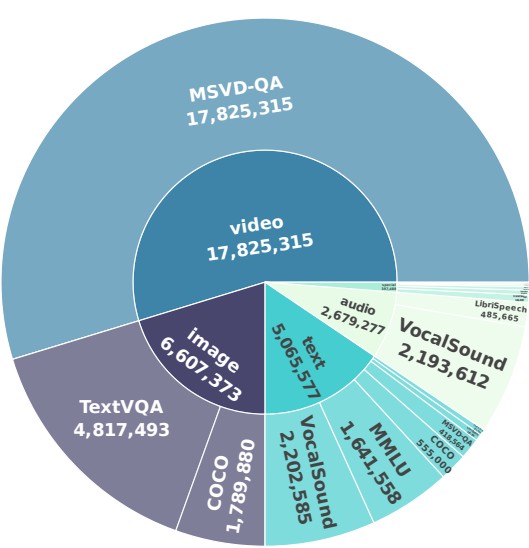

Figure 12: **Token-level count statistics for different modalities.** This image provides a token-level analysis. The inner ring represents different modalities, while the outer ring shows the corresponding datasets. We process each sample's modality components through specific encoders and tokenizers to generate token sets, which are then categorized and summarized in the pie chart. Although the number of samples across datasets is balanced, the token count varies significantly by modality—video encoding produces far more tokens than text encoding, for instance. This underscores the need to compute importance scores within each modality's token set and normalize by token count for fair cross-modality comparisons.

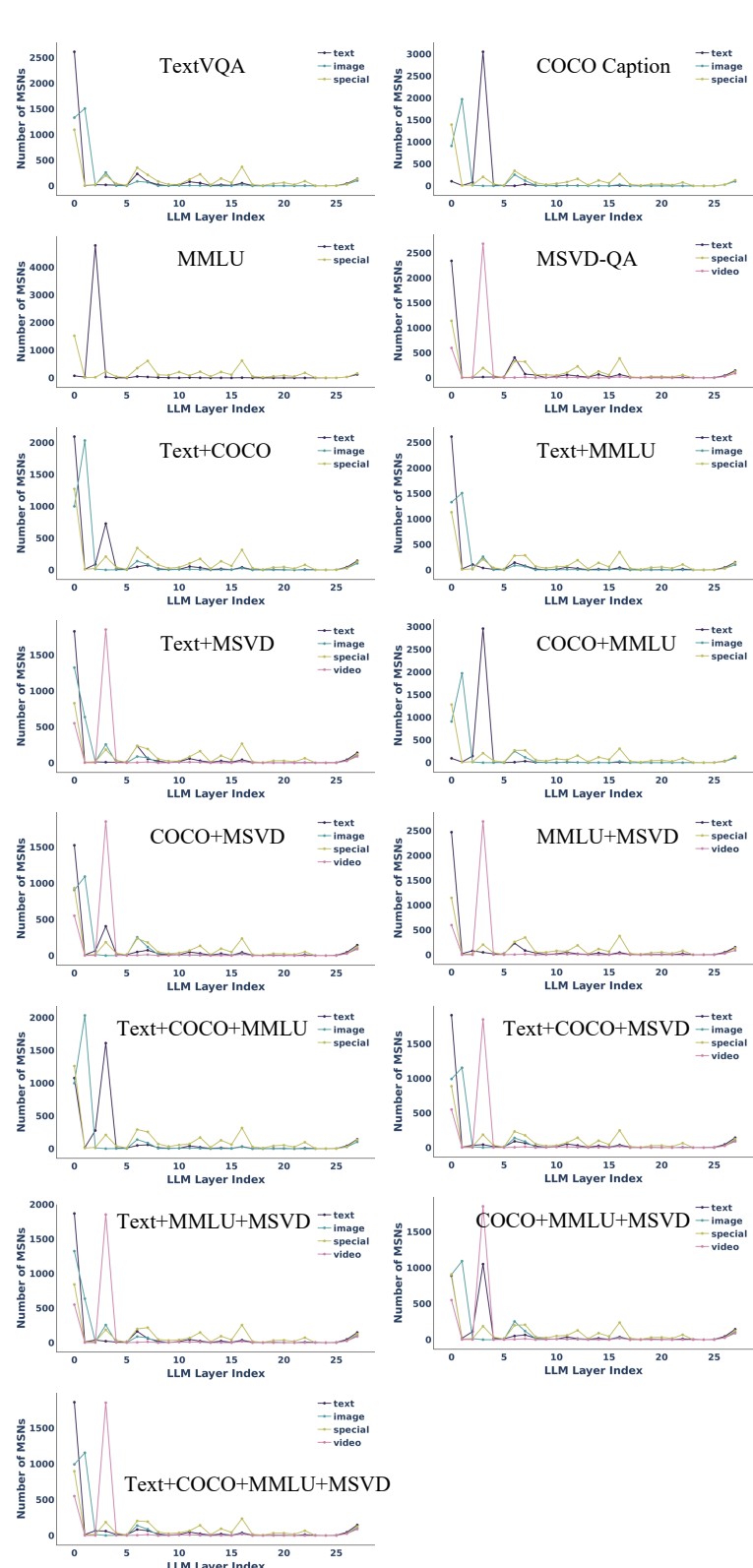

Figure 13: The distribution of modality-specific neurons across different layers, derived from all possible dataset combinations in Qwen2-VL.

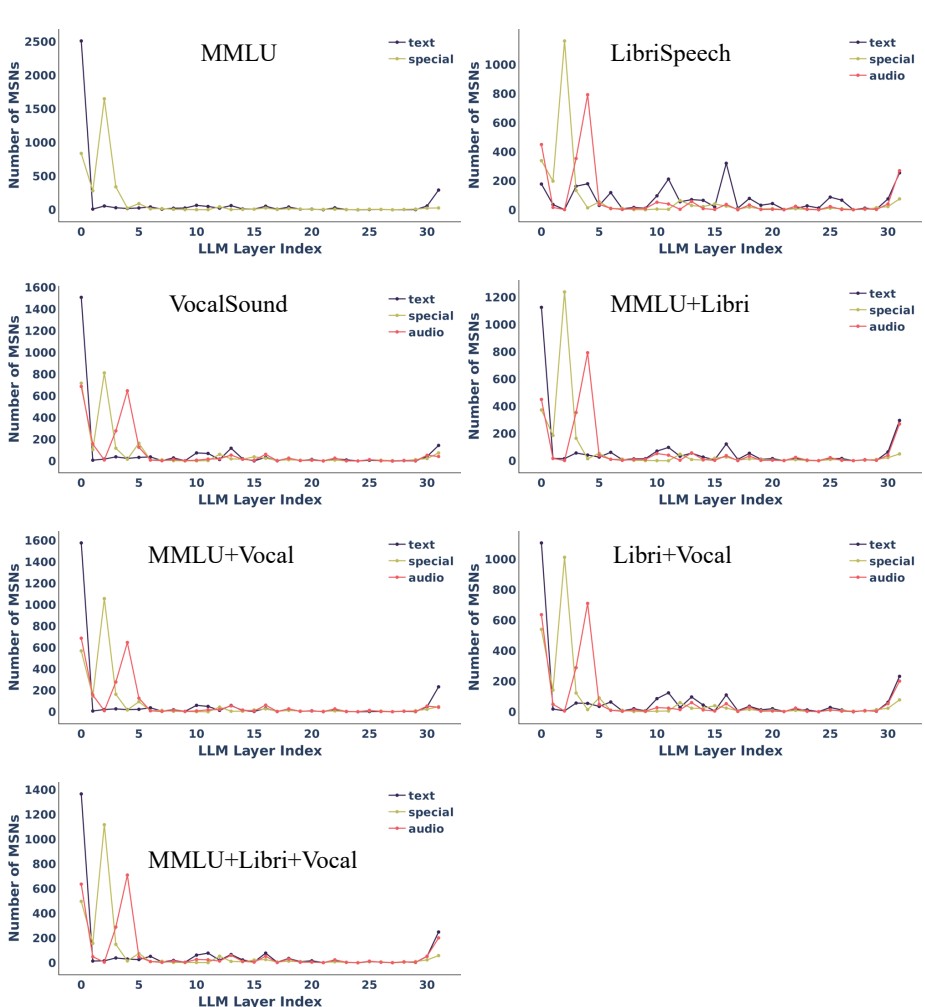

Figure 14: The distribution of modality-specific neurons across different layers, derived from all possible dataset combinations in Qwen2-Audio.

