# OpenReview forum: "MINER: Mining the Underlying Pattern of Modality-Specific Neurons in Multimodal Large Language Models"
_ICLR.cc/2025/Conference — Submitted to ICLR 2025_

### Official Review · Reviewer_FDaH · 2024-10-15

**Soundness:** 3
**Presentation:** 3
**Contribution:** 3
**Rating:** 6
**Confidence:** 3

**Summary:**

This paper introduces a novel concept of modality-specific neurons (MSNs) within multimodal large language models (MLLMs).  These MSNs are specialized neurons that play a crucial role in processing information from a particular modality, such as image or text. The authors propose a framework named MINER to identify and analyze these MSNs, utilizing a unique token-level analysis pipeline to gauge the importance of neurons for specific modalities. This approach diverges from traditional sample-level methods and offers a more fine-grained understanding of how MLLMs process multimodal information.

Through extensive experimentation on a variety of MLLMs and multimodal benchmarks, the authors demonstrate the significant impact of MSNs on model performance. Notably, they show that even a small reduction in MSNs can lead to noticeable performance drops, highlighting the critical role these neurons play. The paper also reveals intriguing phenomena such as "semantic probing," where audio modalities seem to reach out to text modalities, and "semantic telomeres," where special tokens anchor themselves to text modalities.

**Strengths:**

1. This paper is the first to investigate the concept of modality-specific neurons (MSNs) in MLLMs.  This is a significant and novel contribution that opens up a new avenue of research in the field of interpretability and explainability for multimodal models.

2. This approach is shown to be effective in identifying MSNs that significantly impact model performance.  For example, deactivating just 2% of the MSNs identified by MINER significantly reduce MLLMs performance (0.56 ∼ 0.24 ↓ for Qwen2-VL, 0.69 ∼ 0.31 ↓ for Qwen2-Audio).

3. The authors uncover some impactful trends such as finding that different modalities tend to converge in the lower layers of the MLLM.  They also identify two phenomena, "semantic probing" and "semantic telomeres", which describe how audio and special modality tokens, respectively, interact with text modality tokens.

**Weaknesses:**

1. A potential weakness of the paper lies in its approach to modality separation. The authors acknowledge the challenge of completely separating modalities due to the attention mechanism blending information across modalities. They hypothesize that information largely remains within its modality set as it passes through the LLM layers, with minimal transfer between modalities. They assume the attention mechanism functions only within distinct token sets, preventing information exchange between modalities. However, this assumption oversimplifies the attention mechanism, which is designed to capture dependencies regardless of modality. By limiting information flow, the authors may artificially constrain the model.

2. The authors restrict their analysis to neurons in the FFN module of the MLLMs, based on prior work highlighting the role of FFN in knowledge encoding. However, other modules, such as the attention layers, might also contain modality-specific neurons or contribute to modality-specific processing in ways not captured by this analysis.

3. The experiments primarily focus on image and text or audio and text combinations. Exploring more diverse combinations, such as image and audio or all three modalities together, could reveal a more complete picture of MSN interactions.

**Questions:**

See weaknesses.

---

### Official Review · Reviewer_UaFq · 2024-10-28

**Soundness:** 3
**Presentation:** 3
**Contribution:** 3
**Rating:** 5
**Confidence:** 3

**Summary:**

The paper proposes a method (MINER) for identifying modality-specific neurons (MSNs) in multimodal LLMs. There are four stages to this process, and it is meant to improve the explainability (or transparency) of these large models. As few as 2% of neurons seem to play a key role in the multimodal connections. The authors invoke telomeres (!) to explain how modality-specific neurons interact across layers.

**Strengths:**

- The MINER framework is relatively new, although information theoretic (e.g., mutual information-based) methods for probing arbitrary multimodal networks already exists so this is not as fundamentally revolutionary as it is evolutionary.
- The experiments are extensive and detailed, with Qwen2-VL and Qwen2-Audio models and multiple datasets.
- The identification of "semantic probing" is not entirely new but “semantic telomeres” may be, potentially opening new avenues in XAI methods and may be generalizable across different modalities. It is highly advised to rename ‘telomeres’, though.

**Weaknesses:**

- The model should apply to modalities other than vision and audio, but the evaluation does not extend beyond these modalities. This is relatively minor, of course, since a paper that introduces a new method is not required to exhaust all empirical possibilities.
- The "modality separation" approach assumes minimal cross-modal information flow, which is a major oversimplification. It is not clear from an initial review to what extent removing this brick from the base of the theoretical structure collapses the rest. To a large extent this weakness is touched on in the first limitation in Sec 6.1, but it is not _addressed_. The natural interdependence between modalities, throughout the layers, should be better explained or explored. This is a more major limitation.
- It would be preferable to connect the concepts of ‘semantic telomeres’, for example, to real-world problems. What is the practical impact of this work?
- Other interpretability techniques for multimodal LLMs are not evaluated whatsoever.

**Questions:**

- Could you expand some of the terminological definitions, especially "semantic probing" and "semantic telomeres”?
- Could you compare MINER to other neuron-selection frameworks (including from unimodal domains) and clarify where your method reduces perceived gaps in benchmarking or explainability?
- Would it be possible to improve the legibility of all figures (especially Figs 5 and 6)

---

### Official Review · Reviewer_3wuL · 2024-10-29

**Soundness:** 2
**Presentation:** 3
**Contribution:** 3
**Rating:** 5
**Confidence:** 4

**Summary:**

The paper presents **MINER**, a novel framework designed to identify Modality-Specific Neurons (MSNs) within multimodal large language models (MLLMs). Through a structured, four-stage process, MINER attempts to localize neurons specific to each modality, aiming to uncover patterns in how different modalities interact within the MLLM framework. Key claims include the framework’s ability to select MSNs that, when deactivated, result in substantial performance drops, suggesting their importance in maintaining multimodal functionality.

**Strengths:**

- **Clarity and Organization**: The paper is well-structured, with a clear flow that makes complex concepts accessible.
- **Novelty**: The introduction of a framework specifically for modality-specific neuron detection in multimodal models is innovative.
- **Interesting Observations**: The identified phenomena, particularly semantic probing and semantic telomeres, add valuable insights to MLLM research.
- **Thorough Ablation studies**: The ablation studies showcase the impact of different strategies when various importance metrics are employed.

**Weaknesses:**

- **Validation of MSN Specificity**: The evidence provided to confirm that the identified neurons are genuinely modality-specific (i.e., MSNs) is insufficient.
- **Implications Remain Unclear**: The practical implications and potential applications of identifying MSNs are not well-discussed, limiting the paper’s impact.
- **Data Presentation and Clarity**: Several figures, particularly Figure 6, are challenging to interpret due to their small size and lack of clear labels.

**Questions:**

### **Detailed Comments and Suggestions for Improvement**

**Major Comments**

1. **Insufficient Validation of Modality-Specific Neurons (MSNs)**
   The deactivation experiments indicate that identified neurons are crucial for model performance, but this does not confirm they are modality-specific. If a neuron is equally essential across all modalities, its deactivation would similarly result in performance degradation without necessarily being MSN. Additional experiments should be introduced to verify the modality specificity of these neurons.

2. **Need for Cross-Modality MSN Overlap Analysis**
   Given the inherent interactions between modalities, it would be valuable to analyze and present the extent of overlap among MSNs across different modalities. Showing the percentage of MSNs significant to multiple modalities and their distribution across layers would clarify the degree of cross-modality dependency.

3. **Lack of Practical Implications for MSN Discovery**
   The discovery of MSNs is conceptually valuable, but the paper lacks a discussion on how these findings could practically benefit MLLM design or solve current challenges. Possible applications would increase the paper’s practical relevance.

4. **Potential Plagiarism in Figure 1**
   Figure 1 (left) closely resembles Figure 1 of (Huo et al., 2024) but lacks proper citation. This raises concerns about originality.

5. **Explanation Gaps for Key Concepts**
   Terms like **semantic probing** and **semantic telomeres** are introduced in the introduction without clear definitions, which can confuse readers. Providing definitions for these terms at the outset would improve clarity.

6. **Issues with Experiment Design and Metrics**
   - **Typo**: There’s a typographical error on page 3, line 155 (“={text.”), which likely should be “={text, image}.”
   - **t-SNE vs. UMAP**: The paper utilizes t-SNE to compare embeddings across layers, but given t-SNE's stochasticity, UMAP could offer more consistent comparisons across layers, maintaining better global structure.
   - **Inconsistent Table Order**: Table 3 appears before Table 2, disrupting the flow and causing potential confusion.

**Minor Comments**

1. **Explanation for Confirmatory Experiment 5.5 (Hypothesis 1)**
   Experiment 5.5 is referenced as supporting Hypothesis 1, but the link between the experiment’s findings and the hypothesis lacks clarity. A more detailed explanation would strengthen this connection.

2. **Typographical Issues in Figures and Captions**
   - **Figure 6 Text Size**: The text in Figure 6 is too small, making it difficult to interpret. Larger font sizes would enhance readability.
   - **Axis Titles Missing in Figure 6(a)**: Figure 6(a) lacks x-axis titles, complicating interpretation. Clear labeling of axes would improve understanding.
   - **Overcrowded Plot in Figure 6(a)**: Multiple experimental results are displayed within a single figure, which is unconventional and may confuse readers. Separating these plots or including distinct legends for each result would improve clarity.
   - **Inadequate Captions for Figures 6(c), 6(d), and 6(e)**: These figures lack proper captions, making it challenging to understand the displayed results without referring back to the main text.

3. **Potential Typographical Error in Table 4**
   In Table 4, the result for “Adaptive” under “Mean 1/2 and Max 1/2” is listed as 0.17, which appears unusually low and could be a typo. Clarifying this result would help in understanding the significance of the presented values.

4. **Experiment Suggestion: Varying MSN Deactivation Levels**
   It would be informative to present results for different levels of MSN deactivation (from 0.5% to 5% at 0.5% intervals). This could reveal the relationship between neuron deactivation levels and performance, providing further insights into the impact of MSNs.

---

### Official Review · Reviewer_fTci · 2024-10-30

**Soundness:** 3
**Presentation:** 2
**Contribution:** 2
**Rating:** 5
**Confidence:** 3

**Summary:**

The authors introduce a novel method, named MINER, to identify Modality-Specific Neurons in Multimodal Large Language Models (MLLMs). This approach comprises four main stages: 1. **Modality Separation**: The method begins by assuming that information within each modality's token set predominantly stays within that set, implying that cross-attention between modalities plays a minor role in MLLM tasks. 2. **Importance Score Calculation**: Instead of assessing the importance of individual activations for each sample, the method aggregates activations across tokens sets within a modality. The authors propose five distinct aggregation techniques for this step. 3. **Sample-Level Aggregation**: These aggregated importance scores are combined to generate a sample-level importance score. 4. **Neuron Selection**: Finally, neurons are ranked based on their importance scores, allowing for the selection of modality-specific neurons. This method offers a systematic approach to pinpoint neurons that are critical to individual modalities within MLLMs, advancing our understanding of how these models handle multimodal information.

The authors should the efficacy for their method on multiple tasks and multiple datasets.

**Strengths:**

The proposed methodology has several notable strengths:

1. It explores the relatively unexplored area of identifying modality-specific neurons in Multimodal Large Language Models (MLLMs), offering insights into the internal workings of these complex models.

2. The authors provide comprehensive results across multiple MLLM architectures and datasets, accompanied by extensive ablation studies and in-depth discussions, demonstrating the robustness and applicability of their approach.

**Weaknesses:**

In my view, there are several weaknesses in the paper's approach:

1. **Modality Separation Assumption**: The assumption of modality separation, as defined in the paper, is overly generalized and lacks robustness.

- **Dataset and Task Specificity**: Modality separation can only be reasonably argued for certain datasets and tasks. For example, it might be valid for specific questions in Visual Question Answering (VQA) but is unlikely to hold across all question types. Moreover, it seems especially problematic to assume modality separation for captioning tasks and datasets, where contextual understanding of the entire image is essential.

- **Dependence on Output Context**: Another critical consideration is whether modality separation should be treated as a function of the output. In captioning tasks, the entire image often correlates with the output, as generating captions requires broader contextual information. In contrast, the input (or prompt) for caption generation may hold less significance, further challenging the assumption of modality separation. Could you, if possible, conduct an ablation study to assess the significance of the output modality in the importance calculation?

- **Question on Importance Calculation**: This brings up a key question regarding the calculation of neuron importance: is the importance score determined solely based on the input, or does it also consider the model's output? If not, would including the model’s output in this calculation lead to different importance scores?

2. *Missing Definition*:  H_L is not defined in section 3.2. No reference is present in any of the equation 2 or 3 on the same page.


3. **Dataset Size Impact**: Table 2 illustrates that different datasets yield different neuron importance scores. However, it's not clear if varying the quantity of data within a single dataset also impacts these scores. For example, if using 100% of the dataset identifies important neurons \(N = \{n_1, n_2, n_3, n_4\}\), would using only 50% or 33% of the dataset result in a different set of important neurons, or would it merely be a subset of \(N\)?  If the importance neurons change, are they still importance neurons, or is there any underlying cause which we are not able to see.


4. **Computation Requirements**: The paper lacks an analysis of the computational resources needed to determine neuron importance. Since high-performance compute resources can be limited, an estimate of the required computational effort is essential to evaluate the feasibility of this explainability method.

**Questions:**

**Questions and Suggestions for the Authors:**

1. Could you add a definition for \( H_L \) and clarify what is meant by the “hidden activation function”? This addition would enhance the readability of the paper. Additionally, Table 3 currently appears a page before Table 2, which disrupts the logical flow.

2. Please clarify the implementation details for Stage I (modality separation) in your proposed method. Refer to the outlined weaknesses for specific questions and concerns. Additionally, is there supporting empirical evidence for this assumption in the existing literature?

3. Please consider adding a section detailing the computational requirements of your approach. Address Weakness 4 by including specifics such as GPU hours, memory demands, and a comparison with other commonly available explainability methods. This will provide the readers with a clearer understanding of the computational resources needed for your method.

4. Please conduct an ablation study using datasets of varying sizes to analyze whether the neurons are a subset of the original set.  Refer to Weakness 3.

---

### Official Review · Reviewer_Ch93 · 2024-11-02

**Soundness:** 3
**Presentation:** 1
**Contribution:** 2
**Rating:** 3
**Confidence:** 4

**Summary:**

This paper proposes a framework named MINER to understand the underlying pattern of modality-specific neurons in multimodal large language models (MLLM). The framework mainly consists of 4 components or steps: separate modalities, calculate importance scores, aggregate importance scores, and select modality-specific neurons. Experiments are conducted to analyze the existence and effectiveness of the modality-specific neurons.

**Strengths:**

1. The motivation is meaningful and interesting. With the increasing focus on MLLM, the underlying logic and pattern of each component of MLLM is important, and it is also under-explored in current research.
2. The proposed pipeline is clear and easy to follow. As shown in this paper, the framework mainly consists of 4 components or steps: separate modalities, calculate importance scores, aggregate importance scores, and select modality-specific neurons.
3. The experiments seem solid. Extent experiments are conducted to answer the four questions mentioned in this paper.

**Weaknesses:**

1. The writing of this paper is very confusing, even with very low-level mistakes. It seems the authors do not understand what is important and the meaning of the formulation. For instance, in the lines 203 and 204, what does H mean? There is even no H in the equation. Besides, there are some inconsistent formats, such as fig. xx, and Figure xxx, which makes this paper not well-prepared for submission.
2. Since this paper focuses on FFN of MLLM, it is better to show the structure of the focused FFN, making it clearer to understand what the purpose of this paper is.
3. The analysis in the experiments is confusing. For example, in line 403 and 404, MMLU (0.31)? Also, in line 407, mentioned in Ob3? I do not understand what the paper means.
4. Whether the analysis in this paper depends on specific models or not? If so, is it accurate to claim that it is the characteristic of MLLM? I wonder if the performance of FFN in MLLM plays a similar role.

**Questions:**

1. Whether the analysis in this paper depends on specific models or not? If so, is it accurate to claim that it is the characteristic of MLLM? I wonder if the performance of FFN in different MLLMs plays a similar role.

---

> ### Comment · Reviewer_Ch93 · 2024-11-26
> **Discussion**
>
> Since I have not seen any response from the Authors, I will keep my rating as I mentioned before.

---

### Meta-Review · Area_Chair_1CeM · 2024-12-15

**Metareview:**

The paper proposes MINER, a framework for identifying modality-specific neurons. The paper is well-motivated and has an interesting concept with a novel direction. However, key shortcomings include unclear design choices for the experiments as well as unanswered questions regarding some of the analysis and findings. Additionally, the writing and presentation of the work require further improvement to make things more clear and easier to follow.

**Additional Comments On Reviewer Discussion:**

No rebuttal was provided and the reviewers kept their original scores.

---

### Decision · Program_Chairs · 2025-01-22

Reject